

# Spin liquid versus spin orbit coupling on the triangular lattice

Jason Iaconis[1*], Chunxiao Liu[1], Gábor B. Halász[2] and Leon Balents[2]

**1** Department of Physics, University of California, Santa Barbara, CA 93106-9530, USA
**2** Kavli Institute for Theoretical Physics, University of California,
Santa Barbara, CA 93106-4030, USA

* jiaconis@physics.ucsb.edu

## Abstract

In this paper, we explore the relationship between strong spin-orbit coupling and spin liquid physics. We study a very general model on the triangular lattice where spin-orbit coupling leads to the presence of highly anisotropic interactions. We use variational Monte Carlo to study both $U(1)$ quantum spin liquid states and ordered ones, via the Gutzwiller projected fermion construction. We thereby obtain the ground state phase diagram in this phase space. We furthermore consider effects beyond the Gutzwiller wavefunctions for the spinon Fermi surface quantum spin liquid, which are of particular importance when spin-orbit coupling is present.



# 1 Introduction

Quantum spin liquids (QSLs) are exotic phases of correlated electrons possessing highly entangled ground states, exotic fractionalized excitations, and typically, the absence of any magnetic order [1, 2]. Historically, studies of QSLs focused on spin-rotationally invariant Heisenberg models, but in recent years, strongly anisotropic interactions arising from spin-orbit coupling have come under focus [3]. In the famous Kitaev honeycomb model, bond-dependent interactions lead to an exactly solvable model with a spin liquid ground state [4]. Remarkably, it was later shown that these directional interactions can be generated in real materials when spin-orbit effects are present [5,6]. In turn, this has led to the recent discoveries of many candidate 'Kitaev' materials and has paved the way for the study of spin liquid physics in spin-orbital systems. One recent example of particular interest is the material YbMgGaO$_4$ [7–11]. This system very likely contains directional interactions of significant strength. Moreover, thermodynamic and inelastic neutron scattering measurements have been interpreted as supporting a QSL state with a Fermi surface of neutral spin-1/2 excitations, "spinons", in this material.

Spin-orbit generated interactions invariably lead to a strong breaking of spin-rotation symmetry. A consideration of this symmetry in spin liquids can then reveal new and unexpected physics. One striking feature is that the lowered symmetry allows for new distinct spin-liquid phases which do not exist in the rotationally invariant case [12,13]. There exists a systematic method of classifying these phases, given by the so-called projective symmetry group (PSG) [14]. This approach also gives a method for constructing a wave function for each phase, as a Gutzwiller projection of a free fermion state.

We will study a very general spin-orbit coupled model on a triangular lattice which is believed to describe YbMgGaO$_4$ [15–17] and focus specifically on the possibility that this model contains spin liquid physics. We look at the allowed spin liquid phases and use the PSG as a starting point of our analysis. However, our main tool throughout this work is the variational Monte Carlo (VMC) technique. With this numerical technique, one performs Monte Carlo sampling of the quantum wave function in the many-body basis where electrons are localized on each site, allowing one to work with trial states which would otherwise be intractable.

In this paper, we broadly address three points. First, we expound on the relationship between our model and the PSG wave functions. The VMC allows us to quantitatively compare the energies of the different candidate QSL phases. This approach complements recent studies that work with the states phenomenologically [18,19]. We focus on gapless spin liquids with emergent fermionic excitations and highlight the differences between states with isolated Dirac-like quasiparticles and those with a Fermi surface of gapless excitations.

Second, we compare the QSL states to magnetically ordered states, seeking the region of stability of the former ones. We show that a QSL is favored if we allow for second-neighbor interactions, but that spin-orbit effects work to reduce the size of this phase, in agreement

with Ref. [20]. We then go further and show that, if a natural third-neighbor interaction is also included, then the spin liquid phase is energetically competitive, even in the presence of significant spin-orbit interactions.

Finally, we look at how spin-orbit coupling modifies the properties of a QSL, and how this may lead to distinct observables for experiment. We develop a novel method to incorporate modifications beyond the simplest Gutzwiller projected free fermion state into our trial wave function. This method proceeds by calculating many-body corrections order by order in perturbation theory, and sampling these using VMC. We find that this technique is particularly useful for our problem where spin-orbit interactions introduce qualitative differences between the ground state and our trial states. In particular, we study the effect of spin-orbit coupling on the energies of certain trial states and also demonstrate how unique properties of these wave functions appear in the spin structure factor and in thermal transport properties.

The remainder of this paper is structured as follows. In section II, we define the general spin model on the triangular lattice that we study in our work. In section III, we introduce the variational wave functions given by the PSG analysis, which will form the basis for the rest of our discussion. We first calculate the energies of the different candidate spin liquid ansätze using variational Monte Carlo, then allow for the possibility of magnetic order in our simulation, and finally plot the full variational phase diagram for our Hamiltonian. In section IV, we introduce our new method for improving the simple PSG wave functions. We calculate the corrections to the energy and the spin structure factor of the spinon Fermi surface spin liquid state. We also show how the spin-orbit interactions may result in an appreciable thermal Hall conductivity in this system. Finally, in section V, we summarize our results and discuss the relevance of our work to the material YbMgGaO$_4$.

## 2 The model

In many physical systems, the spin and orbital degrees of freedom of the localized electrons are highly entangled. In these cases, when the rotation symmetry is broken by the surrounding crystal structure, the spin-rotation symmetry is broken as well. Superexchange processes then lead to the generation of highly anisotropic terms in the effective spin Hamiltonian. In these strongly spin-orbit coupled systems, lattice symmetry transformations are accompanied by an equivalent transformation in spin space. Following Ref. [15], we consider the Hamiltonian

$$H = H_\pm + H_z + H_{\pm\pm} + H_{\pm z},$$

$$H_\pm = J_\pm \mathsf{H}_\pm = J_\pm \sum_{\langle ij \rangle} \left( S_i^+ S_j^- + S_i^- S_j^+ \right),$$

$$H_z = J_z \mathsf{H}_z = J_z \sum_{\langle ij \rangle} S_i^z S_j^z,$$

$$H_{\pm\pm} = J_{\pm\pm} \mathsf{H}_{\pm\pm} = J_{\pm\pm} \sum_{\langle ij \rangle} \left( \gamma_{ij} S_i^+ S_j^+ + \gamma_{ij}^* S_i^- S_j^- \right),$$

$$H_{\pm z} = J_{\pm z} \mathsf{H}_{\pm z}$$
$$= i J_{\pm z} \sum_{\langle ij \rangle} \left[ (\gamma_{ij}^* S_i^z S_j^+ - \gamma_{ij} S_i^z S_j^-) + (i \leftrightarrow j) \right],$$

(1)

where $\gamma_{ij} = 1, e^{2\pi i/3}, e^{-2\pi i/3}$ for bonds $\langle ij \rangle$ along the $\vec{a}_1, \vec{a}_2, \vec{a}_3$ directions, respectively (see Fig. 1a)). This is the most general nearest-neighbor Hamiltonian which is invariant under the symmetry generators of the system: the translations $\mathcal{T}_{1,2}$ along the $\vec{a}_{1,2}$ directions, the sixfold

roto-reflection $\mathcal{S}_6$ within the plane of the lattice, the twofold rotation $\mathcal{C}_2$ around a bond in the $\vec{a}_3$ direction, and time reversal $\Theta$. (Note that the threefold rotation $\mathcal{C}_3 = \mathcal{S}_6^2$ and the inversion $\mathcal{I} = \mathcal{S}_6^3$ are both generated by the sixfold roto-reflection. Conversely, the sixfold roto-reflection $\mathcal{S}_6 = \mathcal{C}_3^2 \mathcal{I}$ is a combination of a 120° rotation and an inversion.) The symmetry generators are all discrete and act simultaneously in real space and spin space. In particular, they transform the coordinates $x_1, x_2$ of a general lattice point $\vec{r} = x_1 \vec{a}_1 + x_2 \vec{a}_2$ as

$$
\begin{aligned}
\mathcal{T}_1 : \quad & (x_1, x_2) \to (x_1 + 1, x_2), \\
\mathcal{T}_2 : \quad & (x_1, x_2) \to (x_1, x_2 + 1), \\
\mathcal{C}_2 : \quad & (x_1, x_2) \to (x_2, x_1), \\
\mathcal{S}_6 : \quad & (x_1, x_2) \to (x_1 - x_2, x_1), \\
\Theta : \quad & (x_1, x_2) \to (x_1, x_2),
\end{aligned}
\tag{2}
$$

while they transform the spin components $(S^x, S^y, S^z)$ as

$$
\begin{aligned}
\mathcal{T}_{1,2} : \quad & (S^x, S^y, S^z) \to (S^x, S^y, S^z), \\
\mathcal{C}_2 : \quad & (S^x, S^y, S^z) \to (-\tfrac{1}{2}S^x + \tfrac{\sqrt{3}}{2}S^y, \tfrac{\sqrt{3}}{2}S^x + \tfrac{1}{2}S^y, -S^z), \\
\mathcal{S}_6 : \quad & (S^x, S^y, S^z) \to (-\tfrac{1}{2}S^x + \tfrac{\sqrt{3}}{2}S^y, -\tfrac{\sqrt{3}}{2}S^x - \tfrac{1}{2}S^y, S^z), \\
\Theta : \quad & (S^x, S^y, S^z) \to (-S^x, -S^y, -S^z).
\end{aligned}
\tag{3}
$$

Importantly, the Hamiltonian does not generically have a continuous spin-rotation symmetry because the XXZ terms $\mathsf{H}_\pm$ and $\mathsf{H}_z$ break the $SU(2)$ spin symmetry down to an in-plane $U(1)$ spin symmetry, while the remaining terms $\mathsf{H}_{\pm\pm}$ and $\mathsf{H}_{\pm z}$ further break the $U(1)$ spin symmetry down to discrete spin symmetries that are intertwined with appropriate lattice symmetries.

It is helpful to write the $\mathsf{H}_{\pm\pm}$ and $\mathsf{H}_{\pm z}$ terms in a slightly different form to further expose the symmetries:

$$
\begin{aligned}
\mathsf{H}_{\pm\pm} &= \sum_{\langle ij \rangle} (\gamma_{ij} S_i^+ S_j^+ + \text{h.c.}) \\
&= 4 \sum_{\langle ij \rangle} \left[ (\vec{S}_i \cdot \hat{n}_{ij})(\vec{S}_j \cdot \hat{n}_{ij}) - \tfrac{1}{2}(S_i^x S_j^x + S_i^y S_j^y) \right], \\
\mathsf{H}_{\pm z} &= \sum_{\langle ij \rangle} \left[ (i\gamma_{ij} S_i^+ S_j^z + \text{h.c.}) + (i \leftrightarrow j) \right] \\
&= 2 \sum_{\langle ij \rangle} \left[ \{ (\vec{S}_i \times \hat{n}_{ij}) \cdot \hat{z} \} S_j^z + (i \leftrightarrow j) \right].
\end{aligned}
\tag{4}
$$

where $\hat{n}_{ij}$ is the unit vector pointing from site $i$ to site $j$. The term $\mathsf{H}_{\pm\pm}$ has a 'clock' structure where spins would like to align along the 120° bond directions, and the term $\mathsf{H}_{\pm z}$ also has a bond dependent structure that incorporates the $\hat{z}$ direction.

There are several cursory reasons one may expect to find spin liquid physics in this model. For one, due to its strong frustration, the triangular lattice has a long and storied history as a spin liquid candidate [21–25]. Beyond that, the form of the anisotropic part of $H$ is highly reminiscent of the interactions in the Kitaev honeycomb model [4], where the direction-dependent spin-spin interactions frustrate the coupling in a way which renders all magnetic orders energetically unfavorable.

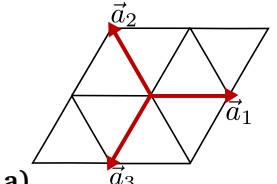 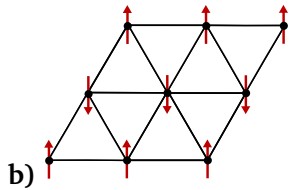 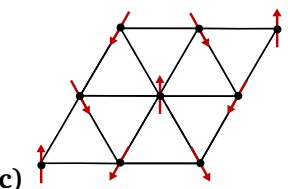

Figure 1: **a)** The three lattice bonds $a_1$, $a_2$, and $a_3$. The two commensurate magnetic orders we consider are **b)** stripe order and **c)** 120° antiferromagnetic order.

## 3 Spin liquid wave functions

### 3.1 Generalities of parton wavefunctions

The ground state wave function in a quantum spin liquid is completely symmetric under all the symmetries of the Hamiltonian. The PSG gives a systematic classification of the allowed spin liquid phases under such a set of symmetries [14]. In the process, it also gives a construction of a representative wave function for each phase. It is a surprising fact that, in many cases, the number of allowed spin liquid phases increases as the symmetry is reduced [12,13]. Spin liquids are fundamentally defined by their fractionalized quasiparticle excitations, whose behavior can be described phenomenologically by a mean-field Hamiltonian. The PSG classifies the fractionalized symmetry by identifying the allowed form of the mean-field Hamiltonians. In general, these excitations can realize the symmetries of the original Hamiltonian in a non-trivial manner.

One starts by writing the physical spin operator $\vec{S}_i$ in terms of fermionic parton operators:

$$\vec{S}_i = \frac{1}{2} f_{i\alpha}^\dagger \vec{\sigma}_{\alpha\beta} f_{i\beta}. \tag{5}$$

The parton operators $f_{i\sigma}, f_{i\sigma}^\dagger$ live in a larger Hilbert space than the spins $S_i$. To remedy this, one must also include the strict gauge constraint on the allowed states:

$$\sum_\sigma f_{i\sigma}^\dagger f_{i\sigma} = 1. \tag{6}$$

In this paper, we enforce Eq. (6) at the level of the wave function. This is accomplished by applying the Gutzwiller projection operator $\mathscr{P}$ to a state $|\psi_0\rangle$ in the fermionic space:

$$|\Psi\rangle = \mathscr{P}|\psi_0\rangle,$$
$$\mathscr{P} = \prod_i n_i(2 - n_i). \tag{7}$$

The projected wave function $|\Psi\rangle$ lives in the proper Hilbert space of spins and, with a suitable choice of $|\psi_0\rangle$, is highly entangled in real space. Furthermore, with some minor improvements, such an ansatz can be made to give variational energies which are competitive with the most state of the art 2D DMRG calculations [23].

For the state $|\psi_0\rangle$, we take a "mean field" wavefunction, which is the ground state of some quadratic fermion Hamiltonian. The parameters of that fiduciary Hamiltonian then become variational parameters in the ansatz. When the fermions are allowed to hop in the mean field Hamiltonian, the partons become deconfined in the corresponding spin liquid phase. In

general, the quadratic mean-field Hamiltonian can be written as

$$\mathcal{H}_{mf} = \sum_{i,j,\alpha} \mathrm{Tr}\left[\sigma^\alpha \Phi_i u_{ij}^\alpha \Phi_j^\dagger\right], \tag{8}$$

$$\Phi_i = \begin{pmatrix} f_{i\uparrow} & f_{i\downarrow}^\dagger \\ f_{i\downarrow} & -f_{i\uparrow}^\dagger \end{pmatrix}, \tag{9}$$

where $\alpha = 0, x, y, z$. A local gauge transformation, such as $f_{i\sigma} \rightarrow e^{i\theta_i \sigma^z} f_{i\sigma}$, changes $H_{mf}$ but leaves the physical spin operator $\vec{S}_i$ unchanged. Since the physical wave function is unchanged, all mean-field Hamiltonians related by such local gauge transformations must be equivalent. The parton Hamiltonian $\mathcal{H}_{mf}$ can therefore ostensibly break the symmetries of $H$ as long as there exists a local gauge transformation which restores the symmetry. In this case, we say that the quasiparticle realizes the symmetry nontrivially. The role of the PSG is to determine the set of allowed mean-field Hamiltonians which cannot be connected to each other by such a gauge transformation. Importantly, $\mathcal{H}_{mf}$ is always invariant under some global transformations $\Phi \rightarrow \Phi \cdot W$, where $W \in G$. The group $G \supseteq \mathbb{Z}_2$ of such global transformations is known as the 'invariant gauge group' (IGG) and determines the form of the gauge group around which fluctuations of the gauge field may occur. In this work, we consider $U(1)$ spin liquids with IGG $= U(1)$.

A more complete study would also include $\mathbb{Z}_2$ spin liquids (IGG $= \mathbb{Z}_2$). However, even restricting to nearest-neighbor couplings, there are at least 18 different $\mathbb{Z}_2$ mean-field ansätze. To avoid this complexity, we neglect these candidate QSLs for the present work. This is at least consistent with recent work on several related triangular lattice spin systems, for which the $U(1)$ spin liquids have proven to have competitive energies [21, 23]. Furthermore, the spinon Fermi surface QSL suggested by several previous papers for YbMgGaO$_4$ falls into the $U(1)$ class.

## 3.2 Six specific parton states

The PSG classification of $U(1)$ QSLs for the space group of our model was done in Ref. [18]. There are 6 distinct nearest-neighbor mean-field Hamiltonians:

$$\mathcal{H}_{mf}^{(1)} = \sum_{\langle ij \rangle, \sigma} \left[ t_{ij} f_{i\sigma}^\dagger f_{j\sigma} + \mathrm{h.c.} \right], \tag{A1/B1}$$

$$\mathcal{H}_{mf}^{(2)} = i \sum_{\langle ij \rangle} \left[ t_{ij} f_{i\alpha}^\dagger (\vec{\sigma}_{\alpha\beta} \cdot \vec{n}_{ij}) f_{j\beta} + \mathrm{h.c.} \right], \tag{A2/B2}$$

$$\mathcal{H}_{mf}^{(3)} = i \sum_{\langle ij \rangle} \left[ t_{ij} f_{i\alpha}^\dagger \{(\vec{\sigma}_{\alpha\beta} \times \vec{n}_{ij}) \cdot \hat{z}\} f_{j\beta} + \lambda_{ij} f_{i\alpha}^\dagger \sigma_{\alpha\beta}^z f_{j\beta} + \mathrm{h.c.} \right]. \tag{A3/B3}$$

The ground state of each mean-field Hamiltonian defines $|\psi_0\rangle$ for the corresponding type of QSL. We distinguish two versions for each mean-field Hamiltonian $\mathcal{H}_{mf}^{(n)}$, which differ only in the way translation symmetry is realized. In the A states, translation acts in the usual way as $t_{ij} = -1$ for all nearest-neighbor bonds $\langle ij \rangle$. Conversely, in the B states, translation acts nontrivially; this is achieved by setting $t_{ij} = \pm 1$ such that the unit cell is doubled and a $\pi$ flux is thread through every other triangle. In the A1/B1/A2/B2 cases, there are no variational parameters (since the overall scale of the Hamiltonian leaves its ground state unchanged), while in the A3/B3 cases, there is a single variational parameter $\lambda/t$.

We note that, importantly, the spinon band structure determines the physical properties of the wave functions and that it is gapless in all 6 states. This is necessary for a U(1) spin liquid to be stable in two dimensions. We now discuss some aspects of these states.

The (A1) state has no mixing between the up and down spin states. In order to satisfy the constraint $\langle f_i^\dagger f_i \rangle = 1$, the band structure then must contain a large Fermi surface. We refer to this state as the **uniform Fermi surface (uFS)** or **spinon metal** state. Notably, although the microscopic Hamiltonian $H$ has only discrete symmetries, the mean-field Hamiltonian of this uFS state is spin-rotationally invariant. This accidental "emergent SU(2) symmetry" is surprisingly robust, and is not an accident of assuming a nearest-neighbor form for $\mathscr{H}_{mf}$. In fact, the PSG does not allow any spin-dependent terms [which would break SU(2) symmetry] in $\mathscr{H}_{mf}$, even for hoppings of arbitrary distance. The argument for this hinges on the fact that both time-reversal ($\Theta$) and inversion ($\mathscr{I}$) symmetries act trivially in this class. First, the operators which implement these symmetries both involve a complex conjugation, time-reversal by definition and inversion due to a site-exchange which corresponds to a Hermitian conjugation. Then, since spin is even under inversion and odd under time reversal, it is odd under their combination, and so a spin-dependent term with any complex coefficient is forbidden in the presence of such a combined symmetry.

The (B1) state also has no mixing of the spin states, but translations act nontrivially on the spinons. The unit cell is then doubled and the band structure contains two Dirac cones. We therefore refer to this state as the **Dirac spin liquid** state. The uFS and Dirac states are the two $U(1)$ spin liquids that can also occur in rotationally invariant systems. Note, however, that spin-dependent quadratic terms are not generically prohibited in the case of the (B1) state and that they in fact appear at the level of third-nearest-neighbor hoppings.

The (A2) and (B2) states are called the **120° clock spin liquid (ClSL)** and the **120° clock + $\pi$ spin liquid (Cl$\pi$SL)**, respectively. These ansätze do mix the spin flavors and orbital degrees of freedom by including bond dependent hoppings. The band structures in both cases contain protected Dirac cones at the $\Gamma$, $M$, and $K$ points in the Brillouin zone.

The (A3) state, called the **Rashba spin liquid (RSL)**, also has Dirac cones at the $\Gamma$, $M$, and $K$ points when $\lambda = 0$ or $t = 0$, and a gap opens at the $\Gamma$ point for intermediate values of $\lambda/t$. Finally, the (B3) state, called the **Rashba + $\pi$ spin liquid (R$\pi$SL)**, contains 4 bands and a small Fermi surface for intermediate values of $\lambda/t$.

### 3.3 Energetics of PSG wave functions

The PSG method gives us the full set of allowed free fermion wave functions that are invariant under the symmetries of the system once the gauge constraint, Eq. (6), is enforced. It tells us nothing, however, about the energies of these wave functions. The PSG gives us a starting ansatz, but is completely agnostic about which state may actually be the ground state.

One simple way to proceed is to work directly with the single particle wave functions by satisfying Eq. (6) on average: $\frac{1}{N} \sum_{i,\sigma} \langle f_{i\sigma}^\dagger f_{i\sigma} \rangle = 1$. However, such a mean field approach requires an infinite number of approximations, the resulting wave functions do not even live in the proper Hilbert space, and thus it cannot give reliable energy estimates. Instead, we carry out a variational analysis based on the fully projected wavefunctions in Eq. (7). We calculate the variational energy $E_s = \langle \Psi_s | H | \Psi_s \rangle$, where $s$ indicates one of the six QSL ansätze.

The results are highly constrained by how the projective symmetries are implemented in the given mean-field Hamiltonian. In particular, the uniform Fermi surface and Dirac spin liquid states are completely $SU(2)$ invariant, and therefore the expectation values of the $J_{\pm\pm}$ and $J_{z\pm}$ terms vanish in these states. Similarly, while both the 'clock' and 'Rashba' Hamiltonians have some spin-orbit terms, only the Rashba Hamiltonians include spin-orbit terms both within and perpendicular to the $xy$ plane. Consequently, the 'clock' wave functions also yield vanishing expectation values for the $J_{\pm z}$ terms.

We performed a variational Monte Carlo simulation and measured the energies of each of our trial wave functions on finite size lattices for system sizes up to $N = 32 \times 32$ sites. Each mean-field wave function, when projected, gives a different pattern of entangled spins, giving

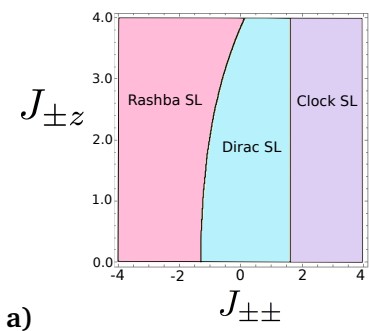
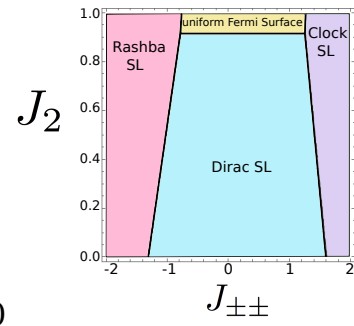

Figure 2: The phase diagram showing only the lowest-energy spin liquid ground states **a)** in the $J_{\pm\pm} - J_{\pm z}$ plane with $J_2 = 0$, and **b)** in the $J_{\pm\pm} - J_2$ plane when $J_{\pm z} = 0$. We set the third neighbor coupling $J_3 = 0$. All energies are measured in units of $J_\pm = 1$. See the main text for a description of the further neighbor terms $J_2$ and $J_3$.

rise to different spin correlations. When $\lambda = 0$, none of the wave functions have any free parameters. Setting $J_\pm = 1$ and scaling to the thermodynamic limit, the corresponding energy densities are then given by

$$
\begin{aligned}
E_{Dirac}/N &= -0.7050(1)[1 + J_z/4], \\
E_{uFS}/N &= -0.4682(5)[1 + J_z/4], \\
E_{Clock}/N &= -0.0645(2) + 0.325(1)J_z - 0.716(1)J_{\pm\pm}, \\
E_{Rashba}/N &= -0.1663(4) + 0.258(1)J_z + 0.741(1)J_{\pm\pm}, \\
E_{Cl\pi}/N &= -0.0619(6) - 0.321(1)J_z - 0.582(1)J_{\pm\pm}, \\
E_{Rsh\pi}/N &= +0.1173(4) + 0.256(1)J_z + 0.525(1)J_{\pm\pm}.
\end{aligned}
\tag{10}
$$

A few observations are apparent. First, we see that the (Cl$\pi$SL) and (R$\pi$SL) ansätze are never competitive energetically in our regimes of interest. While the Dirac state has the lowest energy at $J_{\pm\pm} = 0$, the clock and Rashba spin liquid states become energetically favorable for large positive and negative $J_{\pm\pm}$, respectively. The Rashba states (and only the Rashba states) have an energy which is modified by including $\lambda \neq 0$, which is beneficial only when $J_{\pm z} \neq 0$. In this case, we determine the optimal Rashba state for a given value of $J_{\pm z}$ by numerically minimizing the energy with respect to $\lambda/t$.

The results for a full comparison of energies are presented in Fig. 2a), which shows the state of lowest variational energy amongst the 6 QSLs for all values of $J_{\pm\pm}$ and $J_{\pm z}$. (Note that the phase diagram is qualitatively similar for all values of $J_z$.) Looking ahead, it has been suggested [11] that next-nearest-neighbor interactions may be important in stabilizing a spin liquid ground state for our model. We therefore also looked at the variational energies of our ansätze when XXZ-like next-nearest-neighbor interactions are added (see Eq. 13 in Sec. 3.4.2). In Fig. 2b), we plot the lowest energy states as a function of the next-nearest-neighbor coupling $J_2$ for $J_{\pm z} = 0$. Notice that the Fermi surface state only becomes competitive in energy for very large next-nearest-neighbor coupling.

## 3.4 Magnetic order

### 3.4.1 Parton formulation of ordered states

The PSG wave functions can be used as a starting point on which magnetic order can be added. This is done by adding a site dependent magnetic field $\vec{h}_i$ to the mean-field Hamiltonians,

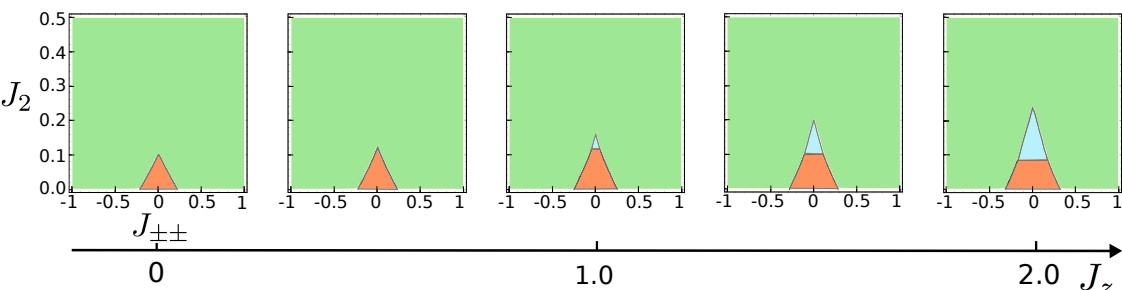

Figure 3: The full $J_{\pm\pm} - J_2 - J_z$ magnetic phase diagram for $J_3 = J_{\pm z} = 0$. Green is stripe order, red is 120° AFM order, and blue is the Dirac spin liquid phase. Stripe order dominates the phase diagram, except for small $J_2$ and $J_{\pm\pm}$. The spin liquid regime also depends strongly on the value of $J_z$ and is greatly reduced when $J_z$ moves away from the isotropic point $J_z = 2$. The horizontal axis on each subplot gives the value of $J_{\pm\pm}$. All energies are measured in units of $J_\pm = 1$.

which define our trial states:

$$\mathcal{H}_{mo} = \mathcal{H}_{mf} - \sum_i \vec{h}_i \cdot \vec{S}_i. \tag{11}$$

Magnetic order can be induced in this way on top of any of the 6 QSL states. In practice, the lowest energies are found by using $\mathcal{H}_{mf}^{(1B)}$, i.e., by perturbing the Dirac spin liquid. Notably, the Zeeman term in this case fully gaps the partons. Consequently, the usual Polyakov argument, which applies to an emergent $U(1)$ gauge theory with fully gapped Dirac fermions in two dimensions, implies that monopole instantons proliferate and the Dirac spinons are confined. Thus, the projected wavefunction built from $\mathcal{H}_{mo}$ describes a state adiabatically connected to a conventional magnetically ordered one.

If $|\vec{h}_i| \to \infty$, Eq. (11) describes classical magnetic order with $|\langle \vec{S}_i \rangle| = 1/2$ on each site. If instead a finite field is used, the value of the magnetic moment can be greatly reduced. In general, the energy should be optimized with respect to the full *set* of Zeeman fields $\vec{h}_i$ on all sites. In practice, such an optimization would have too many parameters. Instead, we guess an appropriate pattern for these fields, and then optimize $|h|/t$ to give the lowest variational energy. For example, in the Heisenberg limit, we choose the field to have a constant magnitude but an orientation with a three-sublattice structure of total vector sum zero (the symmetry pattern of the 120° state):

$$\vec{h}_i = |h|(\cos(\vec{q} \cdot \vec{r}_i + \phi), \sin(\vec{q} \cdot \vec{r}_i + \phi), 0), \tag{12}$$

where $\vec{q}$, $|h|$ and a phase $\phi$ are variational parameters. In this case, the optimal magnetic field of our simple ansatz gives a staggered magnetic moment $|\langle \vec{S}_i \rangle| \approx 0.30$, while the corresponding DMRG calculations for the triangular-lattice Heisenberg model find a staggered magnetic moment $M \sim 0.20$ [24]. Including local correlations in our variational state, for example, by including Jastrow factors, will in general reduce the value of $\langle S \rangle$ further. It is interesting that our PSG analysis provides a general way to construct any ansatz satisfying the constraint of Eq. (6), even allowing us to construct energetically competitive *magnetic* states in addition to giving a general classification of all spin liquid states.

### 3.4.2 Extended model

Implementation of the above method shows that the nearest-neighbor Hamiltonian $H_{nn}$ in Eq. (1) is dominated by magnetic order. To find actual spin liquid physics, we therefore extend

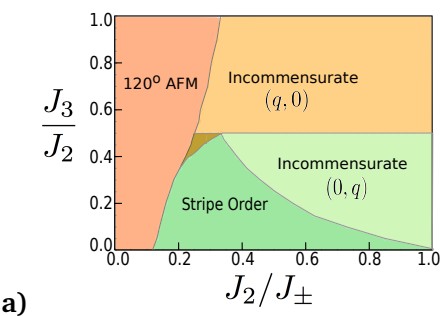
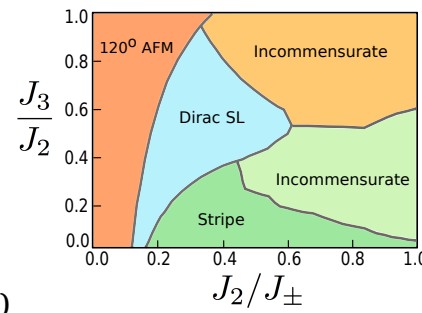

Figure 4: **a)** The classical phase diagram from the Luttinger-Tisza method and **b)** the same quantum phase diagram from variational Monte Carlo at $J_z/J_\pm = 1$ and $J_{\pm\pm} = J_{\pm z} = 0$.

the model to include second- and third-neighbor interactions. Keeping the same relative XY anisotropy, we study the Hamiltonian:

$$H = H_{nn} + J_2 \sum_{\langle\langle ij \rangle\rangle} \left( S_i^+ S_j^- + S_i^- S_j^+ + \frac{J_z}{J_\pm} S_i^z S_j^z \right) + J_3 \sum_{\langle\langle\langle ij \rangle\rangle\rangle} \left( S_i^+ S_j^- + S_i^- S_j^+ + \frac{J_z}{J_\pm} S_i^z S_j^z \right). \quad (13)$$

To avoid complications involving canted magnetic orders, we restrict our attention to the case of $J_{\pm z} = 0$. With this in mind, in this section, we undertake the somewhat ambitious goal of describing the entire four-dimensional phase diagram in terms of the free parameters $J_z$, $J_{\pm\pm}$, $J_2$, and $J_3$, all relative to $J_\pm = 1$.

We first review what is already known about the ground state phase diagram of Eq. (13):

- In the absence of second- and third- neighbor interactions ($J_2 = J_3 = 0$), the Luttinger-Tisza analysis of Ref. [15] tells us the magnetic order when $\vec{S}$ is treated as a classical vector. In that case, there is a phase transition from the 120° staggered antiferromagnetic state [ordered at wavevector $\vec{q}_{120} = (\frac{4\pi}{3}, 0)$] at small $|J_{\pm\pm}|$ to a striped phase [ordered at wavevector $\vec{q}_s = (0, \frac{2\pi}{\sqrt{3}})$] for $|J_{\pm\pm}| \gtrsim 0.25$.

- There is also a great deal of literature on the quantum $J_1 - J_2$ model ($J_{\pm\pm} = J_3 = 0$), in the Heisenberg limit ($J_z = 2J_\pm$) [24, 25]. In this case, growing evidence suggests that a spin liquid phase interpolates between the 120° phase for small $J_2$ and the stripe phase at large $J_2$.

### 3.4.3 VMC results

The advantage of using variational Monte Carlo with simple trial wave functions is that we are able to explore a huge phase space of our Hamiltonian. We consider several ansätze for magnetic order, taking the Zeeman field in the form of Eq. (12) with wavevector $\vec{q}_v = (q, 0)$ or $\vec{q}_v = (0, q)$, where $q$, $|h|$, and a phase $\phi$ are variational parameters, which allows for both commensurate and incommensurate ordering. In practice, we find that the energies of all our ansätze, except for the striped phase with $\vec{q}_s = (0, \frac{2\pi}{\sqrt{3}})$, are independent of $\phi$, even when the $U(1)$ symmetry is broken by $H_{\pm\pm}$. In the stripe phase, we find that the minimum energy is always obtained for $\phi = 0$ when $J_{\pm\pm} > 0$, giving the ordering pattern seen in Fig. 1b), and for $\phi = \pi/2$ when $J_{\pm\pm} < 0$, which rotates all spins by 90°. In Fig. 3, we present our result for the full quantum $J_z - J_{\pm\pm} - J_2$ phase diagram. Notice that our results agree well with the previously understood limits. When $J_2 = 0$, the system acts very similar to the classical case, with a transition between the 120° and stripe orders around $J_{\pm\pm} \approx 0.20 + 0.05J_z$. When a second-neighbor interaction is added, we indeed see that a Dirac spin liquid appears between the 120° and stripe phases. This phase is stable for small $J_{\pm\pm}$, but both large $J_2$ and $J_{\pm\pm}$

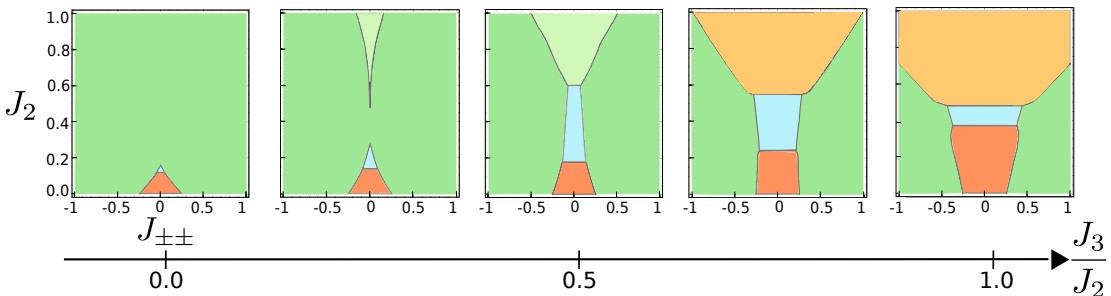

Figure 5: The full $J_{\pm\pm} - J_2 - J_3$ quantum phase diagram for $J_z = 1$ and $J_{\pm z} = 0$. Note that the color scheme is the same as in Fig. 4. Third neighbor interactions $J_3$ strongly disfavor stripe order (dark green) and increase the range of the spin liquid phase (light blue). The horizontal axis on each subplot gives the value of $J_{\pm\pm}$. All energies are measured in units of $J_\pm = 1$.

favor stripe order, leading to the triangular shape of the spin liquid regime which we see in Fig. 3. It is also notable that the extent of the spin liquid phase shrinks dramatically when $J_z$ is lowered from the Heisenberg point. This is in agreement with the DMRG results on this model in Ref. [20].

We are also able to go beyond this model to look at the effect of the third-neighbor XXZ interaction $J_3$. Since both the second- and third-neighbor sites are separated by two lattice bonds, a simple superexchange picture implies that such a term would be present in materials with $J_3 \sim 0.5 J_2$. We will see that the effect of such a term is to enhance the size of the spin liquid regime.

First, we present the results in the classical limit. When $J_{\pm\pm} = 0$, the system has $U(1)$ symmetry and we can solve for the classical magnetic order using the Luttinger-Tisza method since any coplanar magnetic order with a single ordering wave vector will satisfy the hard constraint that $\vec{S}_i = 1/2$ on every site. The result is that, in addition to the usual 120° and stripe phases, $J_3$ favors two additional incommensurate magnetic phases, with ordering wave vectors at $(q, 0)$ and $(0, q)$. These phases can be thought of as the incommensurate versions of the 120° and stripe phases, respectively. A third incommensurate order with wave vector $(q, q)$ also appears classically, but we will ignore this as such a phase never appears in the quantum case. The full classical phase diagram is shown in Fig. 4a) and is independent of $J_z$ since the ordering is always in the $xy$ plane.

Our VMC results on the quantum model agree remarkably well with the classical phase diagram, considering we have used completely different methods. Fig. 4b) shows the results for $J_z = 1.0$. We see that the shapes of the magnetic phases are largely the same as in the classical case, but the intermediate region where the phases meet is occupied by a broad spin liquid regime.

In Fig. 5, we show the full $J_2 - J_3 - J_{\pm\pm}$ phase diagram for $J_z = 1.0$. In addition to the presence of incommensurate magnetic order, the major feature of the data is that the spin liquid regime is enhanced with respect to the $J_3 = 0$ case. The third-neighbor interaction provides further frustration and finds stripe order particularly unfavorable. The spin liquid phase therefore survives to a large value of $J_{\pm\pm}$ when $J_3$ is included.

As mentioned previously, more accurate energies can be found by adding further variational parameters to the wave function, such as allowing for Jastrow factors [26, 27] or performing a small number of Lánczos steps [28]. However, we find that supplementing the PSG wave functions in this way only gives small improvements in the energies, leading to very small shifts of the phase boundaries. In section IV, we look at how we can make *qualitative*

changes to the spin liquid ansätze.

In summary, our variational Monte Carlo calculation allowed us to explore a huge parameter space of the Hamiltonian in Eq. (13) and to obtain quantitative results for the ground state in each parameter regime. When a second-neighbor interaction is added, the *Dirac* spin liquid appears as the ground state between the 120° and stripe phases. This phase shrinks dramatically away from the Heisenberg limit, but is in fact enhanced when a small third-neighbor interaction is included.

# 4 Beyond the PSG wave function

## 4.1 Perturbative correction to the wave function

In this section, we take a more phenomenological approach to studying a quantum spin liquid in the presence of strong spin-orbit coupling. We propose modifications to the mean-field ansätze which can be implemented numerically in the variational wave functions.

The plain mean field ansätze are limited in the amount of complexity they can accommodate. The main issue with the VMC simulation in this context is that the two most energetically competitive states, the Fermi surface and the Dirac spin liquid ones, possess *too* much symmetry. Our trial wave functions have no coupling between the spin and orbital degrees of freedom, which is a feature one would expect to find in the Hamiltonian's true ground state. Furthermore, according to the PSG analysis, no fermion bilinear operators inducing such spin orbit coupling can be added to the uniform Fermi surface Hamiltonian, not even at the further-neighbor level.

Instead, we formulate a method to incorporate *many-body* effects which modify our wave functions. Inspired by the path integral formulation for an interacting quantum field theory, we consider the variational state

$$|\Psi\rangle = e^{-\alpha \mathsf{H}} \mathscr{P} |\psi_0\rangle, \tag{14}$$

where $\mathsf{H} = \mathsf{H}_{\pm\pm}$ is defined in Eq. (1). This form is reminiscent of the Lánczos algorithm, where applications of large powers of an operator project a trial state into the ground state of the given operator. Indeed, if we let $\alpha \to \infty$, this operator projects into the ground state of $\mathsf{H}$. Instead, however, we take a slightly different approach, and let $\alpha$ be a small perturbation on $\mathscr{P}|\psi_0\rangle$, treating $|\Psi\rangle$ as a variational wave function.

There have been previous works combining the Lánczos algorithm with variational Monte Carlo [23, 28]. This proceeds by applying a finite number of Lánczos steps and working with the wave function $|\Psi^{(n)}\rangle = (1 + \sum_{p=1}^{n} \alpha_p H^p)|\psi_0\rangle$, where the series is truncated for some small $n$, and the $\alpha_p$ are left as variational parameters. While this works well if the initial state is very close to the ground state of $H$, it is less effective as a phenomenological tool. The reason is that corrections at any finite order $n$ necessarily scale to zero in the thermodynamic limit. When calculating the *correction* to an expectation value using $|\Psi^{(n)}\rangle$, "disconnected" powers of the Hamiltonian are subtracted off in the numerator, but not in the denominator. The normalization factor in the denominator therefore necessarily grows faster than the numerator with system size. Additional powers of $n$ are then needed to compensate for this fact, but a fully extensive correction is only found at $n \sim N$.

Instead, we have found that the best way to work with the wave function in Eq. (14) numerically is to implement the correction perturbatively in $\alpha$, but to all powers in $n$. To do this, we realize that the expectation value of any operator with respect to our improved wave

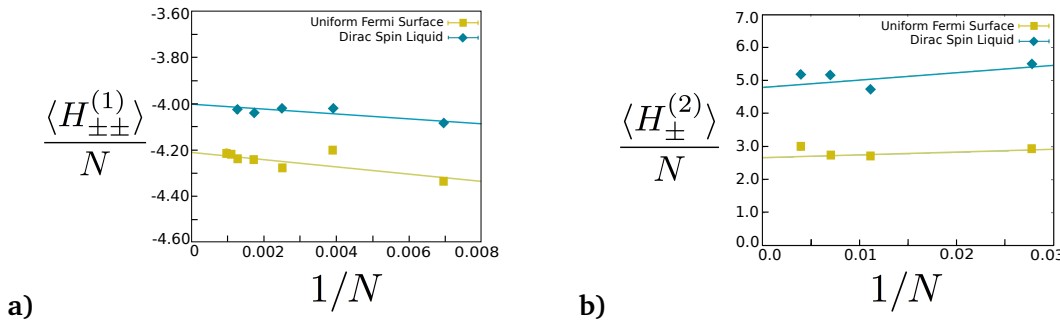

**a)**
**b)**

Figure 6: Finite size scaling of the lowest-order correction to **a)** $\langle H_{\pm\pm} \rangle$ and **b)** $\langle H_\pm \rangle$, for both the uFS (yellow) and Dirac (blue) spin liquid states. The corresponding change in energy is $\Delta E \sim \alpha J_{\pm\pm} \langle H_{\pm\pm} \rangle + \alpha^2 J_\pm \langle H_\pm \rangle$.

function can be written as

$$\langle\!\langle \mathcal{O} \rangle\!\rangle = \frac{\left\langle e^{-\alpha H} \mathcal{O} e^{-\alpha H} \right\rangle_0}{\left\langle e^{-2\alpha H} \right\rangle_0}, \tag{15}$$

where $\langle \cdots \rangle_0$ is the expectation value taken with respect to the unperturbed wave function $\mathcal{P} |\psi_0\rangle$. We use the symbol $\langle\!\langle \cdots \rangle\!\rangle$ to differentiate the theoretical operator expectation values in Eq's (15) and (16) from expectation values we evaluate directly within the VMC simulation.

It is now possible to expand Eq. (15) analogously to diagrammatic perturbation theory. For any Hermitian operator $\mathcal{O}$, the expanded correction reads

$$\langle\!\langle \mathcal{O} \rangle\!\rangle = \frac{\left( \langle \mathcal{O} \rangle_0 - 2\alpha \, \mathrm{Re}[\langle \mathcal{O} H \rangle_0] + \alpha^2 \left( \langle H \mathcal{O} H \rangle_0 + \mathrm{Re}[\langle H^2 \mathcal{O} \rangle_0] \right) + \mathrm{O}(\alpha^3) \right)}{(1 - 2\alpha \langle H \rangle_0 + 2\alpha^2 \langle H^2 \rangle_0 + \mathrm{O}(\alpha^3))}. \tag{16}$$

The subtle difference is that now, by including all powers of $n$, all terms in the denominator exactly cancel the higher order "disconnected" pieces in the numerator. In the VMC calculation, this is expressed by the fact that $\langle H_{ij} H_{k\ell} \rangle \approx \langle H_{ij} \rangle \langle H_{k\ell} \rangle$ as $|(ij) - (kl)| \to \infty$. This way, we are able to measure, in our numerical simulation, many-body corrections to the wave function which survive in the thermodynamic limit. Note that Eq. (16) can not be measured directly within the VMC simulation, but instead can be measured approximately by expanding perturbatively in $\alpha$. As such, the results of our calculation are only variational up to the accuracy of the asymptotic expansion.

In principle, applying the operator $\exp[-\alpha H]$ to our unperturbed trial wave function could cause a phase transition, and we would no longer be working with a spin liquid state. For small $\alpha$, however, we expect that the spin liquid ground state should be stable to such a perturbation. In the spinon metal, in a similar vein to Fermi liquid theory, we expect that these terms only give a correction to the self-energy of spinons near the Fermi surface [29].

## 4.2 Correction to the Energy

To begin, we measure the correction to the energy of the Dirac and uniform Fermi surface states, which arises from including the spin-orbit interaction in our variational wave function. We can directly measure the first and second order corrections numerically.

For any operator $\mathcal{O}$, we write the $n^{th}$ order correction to the expectation value $\langle\!\langle \mathcal{O} \rangle\!\rangle$ from

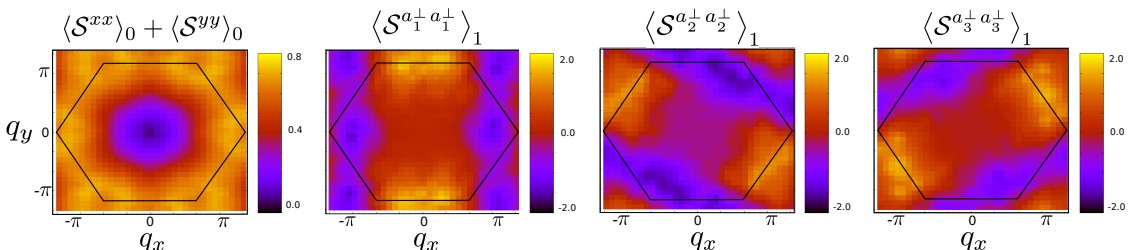

Figure 7: The rotationally invariant spin structure factor (top left) and the perturbative corrections to the spin-polarized structure factors measured with spins pointing perpendicular to the three lattice bond directions $\vec{a}_1, \vec{a}_2$, and $\vec{a}_3$, within the plane of the triangular lattice.

applying $\exp[-\alpha H]$ as $\alpha^n \langle \mathcal{O}^{(n)} \rangle$. Expanding Eq. (16) gives

$$\langle \mathcal{O}^{(1)} \rangle = -2(\text{Re}[\langle H\mathcal{O} \rangle_0] - \langle H \rangle_0 \langle \mathcal{O} \rangle_0), \tag{17}$$

$$\langle \mathcal{O}^{(2)} \rangle = \langle H\mathcal{O}H \rangle_0 + \text{Re}[\langle H^2 \mathcal{O} \rangle_0] - 4\text{Re}[\langle H \rangle_0 \langle \mathcal{O}H \rangle_0]$$
$$- 2\langle H^2 \rangle_0 \langle \mathcal{O} \rangle_0 + 4\langle H \rangle_0^2 \langle \mathcal{O} \rangle_0.$$

In our case, $H = H_{\pm\pm}$ and $\langle H_{\pm\pm} \rangle_0 = \langle H_{\pm\pm} H_\pm \rangle_0 = 0$. Therefore, the spin-orbit part of the Hamiltonian is altered at order $\alpha$, while the rotationally invariant part is corrected at order $\alpha^2$:

$$\langle H_{\pm\pm}^{(1)} \rangle = -2\langle H_{\pm\pm}^2 \rangle_0, \tag{18}$$

$$\langle H_\pm^{(2)} \rangle = \text{Re}[\langle \{H_{\pm\pm}, H_\pm\} H_{\pm\pm} \rangle_0] - 2\langle H_{\pm\pm}^2 \rangle_0 \langle H_\pm \rangle_0,$$

$$\langle H_z^{(2)} \rangle = \text{Re}[\langle \{H_{\pm\pm}, H_z\} H_{\pm\pm} \rangle_0] - 2\langle H_{\pm\pm}^2 \rangle_0 \langle H_z \rangle_0.$$

In Fig. 6, we show the resulting scaling of $\langle H_{\pm\pm}^{(1)} \rangle$ and $\langle H_\pm^{(2)} \rangle$ to the thermodynamic limit. The result is that the spinon metal is more susceptible, compared to the Dirac state, to energetically beneficial corrections to $H_{\pm\pm}$ and less susceptible to detrimental corrections to $H_\pm$ and $H_z$. Putting this together, we find that the optimal value of the variational parameter is $\alpha_{min} \sim J_{\pm\pm}/(J_\pm + J_z)$, which gives an energy correction $\Delta E \sim -J_{\pm\pm}^2/(J_\pm + J_z)$. More precisely, we find that the energy densities after the lowest-order corrections are given by

$$E_{uFS}/N = -0.4682(1 + J_z/4) - \frac{1.56 J_{\pm\pm}^2}{J_\pm + 1.42 J_z},$$

$$E_{Dirac}/N = -0.7050(1 + J_z/4) - \frac{0.84 J_{\pm\pm}^2}{J_\pm + 0.87 J_z}. \tag{19}$$

This implies that that the Fermi surface state becomes energetically superior to the Dirac state between $J_{\pm\pm} = 0.57$ at $J_z = 0$ and $J_{\pm\pm} = 1.54$ at $J_z = 2.0$. One caveat, of course, is that these values of $J_{\pm\pm}$ may fall outside the perturbative regime. Also, while smaller $J_z$ appears to be more favorable for the spinon Fermi surface, this is also the parameter regime which is more susceptible to magnetic order.

## 4.3 Correction to the spin structure factor

Studying the improved variational wave function makes it clear that the spinon metal state in a spin-orbit coupled environment has several unique properties, despite the fact that the mean-field Hamiltonian retains its rotational invariance. Taking our analogy to Fermi liquid theory seriously, the spin-orbit interaction gives a momentum and spin dependent correction

to the self energy. This appears as a momentum dependent correction to the structure factor, which we can again measure directly in our simulation.

We differentiate between the various spin polarized contributions to the spin-spin correlation function:

$$\mathcal{S}^{\alpha\beta}(\vec{q}) = \sum_i e^{i\vec{q}\cdot\vec{r}_i} \langle S_i^\alpha S_0^\beta \rangle. \tag{20}$$

The first-order correction to the correlation function is

$$\langle S_i^\alpha S_j^\beta \rangle_1 = -2 \Big[ \text{Re}[\langle S_i^\alpha S_j^\beta \mathsf{H}_{\pm\pm} \rangle_0] - \langle S_i^\alpha S_j^\beta \rangle_0 \langle \mathsf{H}_{\pm\pm} \rangle_0 \Big]. \tag{21}$$

The results are shown in Fig. 7. The corrections to the spin-polarized structure factor are direction-dependent broad peaks at the $M$ points of the Brillouin zone which appear at order $\alpha \sim J_{\pm\pm}/(J_\pm + J_z)$. Therefore, in a spinon metal with spin-orbit coupling, spin-spin correlations when measured with different spin polarizations are direction dependent. This type of measurement could prove to be an important test to show both the presence of spin-orbit interactions and the absence of spontaneous symmetry breaking. Similar directional peaks can be seen in related models when spin-orbit terms are directly included in the ground state ansatz [30]. We note that these kinds of direction-dependent structure factors have already been measured experimentally by resonant elastic x-ray scattering in the honeycomb lattice iridate $Na_2IrO_3$ [31].

## 4.4 Thermal Hall conductivity

### 4.4.1 General considerations

Thermal transport measurements can be a powerful tool for studying magnetic insulators. The idea is to set up a thermal gradient $\nabla T$ (which is analogous to an electric field) and then measure the heat current $j^{\text{th}}$ in response to it (which is analogous to an electric current). Any heat current in the insulator must be carried by the emergent quasiparticles, giving us a probe of the low energy excitations. The thermal conductivity, $\kappa$, can be defined within linear response as

$$j_\mu^{\text{th}} = -\kappa_{\mu\nu} \partial_\nu T. \tag{22}$$

The spinon Fermi surface QSL is unusual due to the large number of gapless excitations. This leads to a predicted linear $T$ term appearing in the diagonal component of $\kappa$, similar to what one would expect in a metal. The deconfined spinons carry heat in the same way physical electrons carry charge in an electrical conductor. A major difficulty is that many degrees of freedom, most notably phonons, can contribute to the diagonal thermal conductivity, making the measurement challenging.

The thermal *Hall* conductivity, however, given by the off-diagonal component of $\kappa$, should not contain a phonon term. Furthermore, as explained in Ref. [32], it is very difficult to find an effect generated by magnons on the triangular lattice due to a cancellation of the contributions from neighboring edge sharing plaquettes. A large nonzero thermal Hall conductivity could therefore be a strong indicator of exotic physics. Indeed, in Ref. [32], the authors also predict that a spinon metal would display such an effect. However, the reasoning is very subtle, depending on a coupling of the *orbital* motion of the spinons to the external electromagnetic field through the interaction with the internal gauge field.

Here, we argue that there exists a distinct contribution to the thermal Hall conductivity in the spinon metal which is unique to spin-orbit coupled systems and relies only on a Zeeman coupling to the external electromagnetic field. For itinerant fermions with conserved charge,

the presence of spin-orbit coupling can lead to a nontrivial Berry curvature which may induce an anomalous component of the charge Hall conductivity, in the absence of any Lorentz force. This mechanism of anomalous Hall conductivity was explored intensely for Rashba two-dimensional electron gases and in many other models. In the following, we adapt this idea to study the *thermal* conductivity of the Fermi surface QSL state.

The U(1) QSL states studied here have an *emergent* conserved charge, which is the fermion number associated with the emergent U(1) gauge symmetry. Consequently, at the parton level, we can define a current associated with this charge, and we may consider, formally, the emergent conductivity tensor $\sigma_{\mu\nu}^{qp}$ defined with respect to the emergent current and a potential coupling to the associated charge density. This is not the true electrical conductivity, since this is an insulator, and it is also not the thermal conductivity. Thus we proceed in two stages. First, we consider the anomalous emergent Hall conductivity of the spinons. Then, we relate it to the more easily measurable *thermal* Hall conductivity (in principle, the emergent conductivity should also be measureable, but it is not obvious how to do so).

### 4.4.2 Effective quasiparticle Hamiltonian

At the mean field level, the emergent Hall conductivity can be extracted as an integral over the Berry curvature of the occupied spinon bands. Within the simple PSG wave function, the spinon metal is spin-rotationally invariant and therefore has zero Berry curvature. On symmetry grounds, however, we expect that a Hall conductivity should microscopically arise. To estimate it, we consider the 'improved' wave function, and infer a self-energy correction which breaks spin-rotational symmetry and induces a non-zero Berry curvature.

The Berry gauge field (Berry connection) is defined for a single particle system as

$$\vec{A}(k) = -i \langle u_k | \vec{\nabla}_k | u_k \rangle, \tag{23}$$

where $|u_k\rangle$ is defined as in the Bloch wave function. The anomalous Hall conductivity is then given by

$$\sigma_{xy}^{qp} = \oint_{\partial S} \vec{A}(k) \cdot d\vec{k} = \int_S [\vec{\nabla}_k \times \vec{A}(k)] d^2 k, \tag{24}$$

where the first (line) integral is taken around the Fermi surface $\partial S$, while the second (area) integral is taken over the area $S$ spanned by it. This physical quantity is invariant under $U(1)$ gauge transformations, as is immediately evident from its expression in terms of the Berry curvature $\mathscr{B}(k) = \vec{\nabla}_k \times \vec{A}(k)$.

To obtain the Berry curvature, we suppose that the system is described by an effective quasiparticle Hamiltonian including a self-energy correction $\Sigma(k)$ and a Zeeman coupling to an external magnetic field $\vec{B} = h\hat{z}$:

$$\mathscr{H}_{\text{eff}}(k) = \left( f_{k\uparrow}^{\dagger} f_{k\downarrow}^{\dagger} \right) \begin{pmatrix} \varepsilon(k) - h & \Sigma^*(k) \\ \Sigma(k) & \varepsilon(k) + h \end{pmatrix} \begin{pmatrix} f_{k\uparrow} \\ f_{k\downarrow} \end{pmatrix}. \tag{25}$$

We determine the self-energy $\Sigma(k)$ by requiring that the off-diagonal expectation value $\Pi_{\uparrow\downarrow}(k) \equiv \langle f_{k\uparrow}^{\dagger} f_{k\downarrow} \rangle$ calculated using the improved wave function *matches* that calculated using the effective Hamiltonian $\mathscr{H}_{\text{eff}}(k)$.

To proceed, we consider an improved wave function similar to that in Eq. (14):

$$|\Psi\rangle = e^{-\tilde{\alpha}\tilde{H}} \mathscr{P} |\psi_0\rangle, \tag{26}$$

where now we take $\tilde{H} = H_{\pm z}$. The reason for this change is that the previously-considered correction due to $H_{\pm\pm}$ gives exactly zero contribution to $\Pi_{\uparrow\downarrow}$ because it conserves the total

spin $S^z$ modulo 2. The analogous contribution due to $H_{\pm z}$, however, does contribute. We expect that the energetically optimal value of the variational parameter is $\tilde{\alpha} \sim J_{\pm z}/J_0$, where $J_0$ is on the order of the other exchange couplings ($J_\pm$ and $J_z$).

Using the same perturbative expansion as above, the first-order form of $\Pi_{\uparrow\downarrow}(k)$ becomes

$$
\begin{aligned}
\Pi_{\uparrow\downarrow}(k) &= \left\langle e^{-\tilde{\alpha}\tilde{H}} f_{k\uparrow}^\dagger f_{k\downarrow} e^{-\tilde{\alpha}\tilde{H}} \right\rangle_0 \\
&= -\tilde{\alpha}\left( \langle f_{k\uparrow}^\dagger f_{k\downarrow} \tilde{H} \rangle_0 + \langle \tilde{H} f_{k\uparrow}^\dagger f_{k\downarrow} \rangle_0 \right) \\
&\equiv \Pi_R^{(1)}(k) + \Pi_L^{(1)}(k).
\end{aligned}
\tag{27}
$$

If we represent the spin-spin interaction in momentum space with a momentum-dependent form factor

$$
\tilde{\gamma}(k) = \frac{i}{2} \sum_{\mu=1}^{3} \sum_{\pm} \gamma_\mu e^{\pm i \vec{k}\cdot\vec{a}_\mu}, \qquad \gamma_\mu \equiv \gamma_{\vec{0},\vec{a}_\mu},
\tag{28}
$$

the first expectation value $\Pi_R^{(1)}(k)$ takes the form

$$
\begin{aligned}
\Pi_R^{(1)}(k) &= i\tilde{\alpha}\left\langle f_{k\uparrow}^\dagger f_{k\downarrow} \sum_{\langle mn \rangle} \left[ \gamma_{mn} S_m^z S_n^- + (m \leftrightarrow n) \right] \right\rangle_0 \\
&= \frac{\tilde{\alpha}}{N} \sum_{k_1,k_2,k_3} \left[ \langle f_{k\uparrow}^\dagger f_{k\downarrow} f_{k_1\uparrow}^\dagger f_{k_2\uparrow} f_{k_3\downarrow}^\dagger f_{(k_1-k_2+k_3)\uparrow} \rangle_0 \right. \\
&\quad\quad - \left. \langle f_{k\uparrow}^\dagger f_{k\downarrow} f_{k_1\downarrow}^\dagger f_{k_2\downarrow} f_{k_3\downarrow}^\dagger f_{(k_1-k_2+k_3)\uparrow} \rangle_0 \right] \tilde{\gamma}(k_1-k_2) \\
&= -\frac{\tilde{\alpha}}{N} \sum_q \left[ \langle f_{k\uparrow}^\dagger f_{k\uparrow} f_{q\uparrow}^\dagger f_{q\uparrow} \rangle_0 \langle f_{k\downarrow} f_{k\downarrow}^\dagger \rangle_0 \right. \\
&\quad\quad + \left. \langle f_{k\uparrow}^\dagger f_{k\uparrow} \rangle_0 \langle f_{k\downarrow} f_{k\downarrow}^\dagger f_{q\downarrow} f_{q\downarrow}^\dagger \rangle_0 \right] \tilde{\gamma}(k-q),
\end{aligned}
\tag{29}
$$

where we arrive at the last line after conserving spin and momentum in the zeroth-order expectation values as well as using $\tilde{\gamma}(-k) = \tilde{\gamma}(k)$ and $\tilde{\gamma}(0) = 0$.

Performing similar manipulations on $\Pi_L^{(1)}(k)$ and combining the two contributions gives

$$
\Pi_{\uparrow\downarrow}(k) = \Pi_R^{(1)}(k) + \Pi_L^{(1)}(k) = -\tilde{\alpha}\Lambda\tilde{\gamma}(k)\Gamma(k),
$$

$$
\begin{aligned}
\Gamma(k) &= \langle n_{k\uparrow} \rangle_0 \langle 1 - n_{k\downarrow} \rangle_0 + \langle n_{k\downarrow} \rangle_0 \langle 1 - n_{k\uparrow} \rangle_0 \\
&= \coth(h/T)\left[ \langle n_{k\uparrow} \rangle_0 - \langle n_{k\downarrow} \rangle_0 \right],
\end{aligned}
\tag{30}
$$

$$
\begin{aligned}
\Lambda &= \frac{1}{N} \sum_q e^{\pm i\vec{q}\cdot\vec{a}_\mu} \left[ \langle n_{q\uparrow} \rangle_0 + \langle 1 - n_{q\downarrow} \rangle_0 \right] \\
&\sim a^2 \int d^2q \, e^{\pm i\vec{q}\cdot\vec{a}_\mu} \left[ \langle n_{q\uparrow} \rangle_0 - \langle n_{q\downarrow} \rangle_0 \right],
\end{aligned}
$$

where $n_{k\sigma} = f_{k\sigma}^\dagger f_{k\sigma}$ is a number operator and $a = |\vec{a}_\mu|$ is the lattice constant. Importantly, $\Lambda$ is real and independent of both $\mu$ and $\pm$ due to the sixfold symmetry $\mathscr{S}_6$. Furthermore, in the limit of $T \ll |h|$, the integrand is only non-zero in an annulus of thickness $\sim h/(aJ_0)$ around the Fermi surface of radius $\sim 1/a$, and the integral can then be estimated as $\Lambda \sim h/J_0$.

Let us also calculate $\Pi_{\uparrow\downarrow}(k)$ from the effective Hamiltonian in Eq. (25). In the limit of $|\Sigma(k)| \ll |h|$, we obtain

$$
\begin{aligned}
\Pi_{\uparrow\downarrow}(k) &= -\frac{\text{sgn}(h)\,\Sigma(k)}{2\sqrt{h^2 + |\Sigma(k)|^2}}\big[\langle n_{k\uparrow}\rangle_0 - \langle n_{k\downarrow}\rangle_0\big] \\
&= -\frac{\Sigma(k)}{2h}\big[\langle n_{k\uparrow}\rangle_0 - \langle n_{k\downarrow}\rangle_0\big].
\end{aligned}
\tag{31}
$$

Finally, from a comparison of Eqs. (30) and (31), the self-energy in the limit of $T \ll |h|$ becomes

$$
\Sigma(k) = 2|h|\tilde{\alpha}\Lambda\tilde{\gamma}(k).
\tag{32}
$$

The real and imaginary parts of $\Sigma(k)$ are plotted in Fig. 8. Note that the complex phase of $\Sigma(k) \propto \tilde{\gamma}(k)$ winds by $4\pi$ around the $\Gamma$ point.

### 4.4.3 Berry curvature and Hall conductivity

Now we are in a position to calculate the emergent Hall conductivity. First, we rewrite the effective quasiparticle Hamiltonian in Eq. (25) into the standard form

$$
\begin{aligned}
\mathscr{H}_{\text{eff}}(k) &= \big(f_{k\uparrow}^\dagger\, f_{k\downarrow}^\dagger\big)\big[\varepsilon(k)\sigma_0 - h\vec{\beta}(k)\cdot\vec{\sigma}\big]\begin{pmatrix} f_{k\uparrow} \\ f_{k\downarrow} \end{pmatrix}, \\
\vec{\beta}(k) &= \left(-\frac{\text{Re}\,\Sigma(k)}{h}, -\frac{\text{Im}\,\Sigma(k)}{h}, 1\right),
\end{aligned}
\tag{33}
$$

where $|\vec{\beta}(k)| \approx 1$ in the limit of $|\Sigma(k)| \ll |h|$. For such a Hamiltonian, the two bands have Berry curvatures of opposite sign and equal magnitude given by

$$
\begin{aligned}
\mathscr{B}(k) &\sim \vec{\beta}(k)\cdot\big[\partial_{k_x}\vec{\beta}(k)\times\partial_{k_y}\vec{\beta}(k)\big] \\
&\sim \frac{1}{\rho_k}\big\{\vec{\beta}(k)\cdot\big[\partial_{\rho_k}\vec{\beta}(k)\times\partial_{\varphi_k}\vec{\beta}(k)\big]\big\} \\
&\sim \frac{1}{h^2\rho_k}\,\text{Im}\big[\partial_{\rho_k}\Sigma^*(k)\,\partial_{\varphi_k}\Sigma(k)\big],
\end{aligned}
\tag{34}
$$

where we use polar coordinates defined by $k_x = \rho_k\cos\varphi_k$ and $k_y = \rho_k\sin\varphi_k$. Due to the $4\pi$ phase winding of $\Sigma(k)$ (see Fig. 8), there is a finite azimuthal derivative $\partial_{\varphi_k}\Sigma(k) \sim i\Sigma(k)$. From $\partial_{\rho_k}\Sigma^*(k) \sim a\Sigma^*(k)$, the Berry curvature at radius $\rho_k \sim 1/a$ is then on the order of

$$
\mathscr{B}(k) \sim \frac{a^2|\Sigma(k)|^2}{h^2} \sim \tilde{\alpha}^2 a^2\left(\frac{h}{J_0}\right)^2.
\tag{35}
$$

Next, in terms of the Berry curvatures $\pm\mathscr{B}(k)$ of the two bands, the emergent Hall conductivity takes the form

$$
\sigma_{xy}^{qp} = \int d^2k\,\mathscr{B}(k)\big[\langle n_{k\uparrow}\rangle_0 - \langle n_{k\downarrow}\rangle_0\big].
\tag{36}
$$

In the limit of $T \ll |h|$, the integrand is only non-zero in an annulus of thickness $\sim h/(aJ_0)$ around the Fermi surface of radius $\sim 1/a$, and the Hall conductivity can then be estimated as $\sigma_{xy}^{qp} \sim \tilde{\alpha}^2(h/J_0)^3$.

Finally, by virtue of the Wiedemann-Franz law that relates the emergent and the thermal conductivities, the quasiparticle contribution to the thermal Hall conductivity is on the order of

$$
\kappa_{xy} \sim T\sigma_{xy}^{qp} \sim \tilde{\alpha}^2 T\left(\frac{h}{J_0}\right)^3 \sim \frac{Th^3 J_{\pm z}^2}{J_0^5}.
\tag{37}
$$

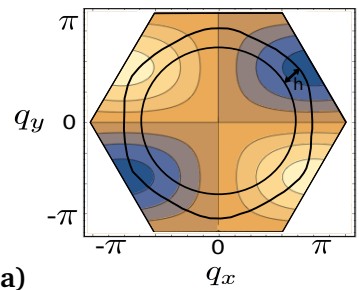 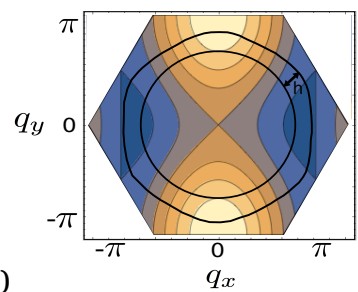

a)                                  b)

Figure 8: The **a)** real and **b)** imaginary components of $\Sigma(k)$ in a magnetic field $\vec{B} = h\hat{z}$. Lighter (darker) contours are positive (negative) contributions. The positions of the spin up and spin down Fermi surfaces in the presence of a nonzero Zeeman field $\propto h$ are also shown.

Interestingly, $\kappa_{xy}$ is proportional to the third power of the magnetic field. Note, however, that this result is valid for a relatively large field ($T \ll |h| \ll J_0$). For a small field ($|h| \ll T \ll J_0$), the factor $\coth(h/T)$ in Eq. (30) contributes an additional factor $\sim (T/h)^2$ to $\kappa_{xy}$, which is then linearly proportional to the magnetic field.

# 5 Discussion

## 5.1 Relationship to other work

In this paper, we have provided a comprehensive commentary on the possibility of spin liquid physics in a very general spin-orbit coupled model on the triangular lattice. In the process, we have attempted to consolidate several previous results on this topic. We began by looking at the $U(1)$ PSG wave functions derived in Ref. [18]. Instead of working with these wave functions phenomenologically, we go beyond their simple mean-field analysis and find quantitative estimates of the energies of these ansätze using variational Monte Carlo.

We also use VMC to give a complete picture of magnetic order in our model. Our results improve on the classical magnetic phase diagrams presented in Refs. [15, 16]. In those works, a phase transition between the 120° and stripe phases is found in the nearest-neighbor model, and it is conjectured that large spin fluctuations may lead to the presence of a nonmagnetic phase. In our work, by building on the PSG ansätze, we also find a phase transition between the two magnetic phases in a similar parameter regime. We further find that second-neighbor interactions are necessary to create a spin liquid ground state and we identify the Dirac spin liquid as the lowest energy state. This confirms and extends earlier studies of the isotropic Heisenberg model [23].

The only other calculation of the full quantum phase diagram in this model was given by the DMRG analysis in Ref. [20]. Our phase diagram agrees with the DMRG analysis when second-neighbor interactions are included. The XXZ anisotropy and $J_{\pm\pm}$ interactions both work to limit the spin liquid phase to a very small region of parameter space. However, we go beyond this and also include a third-neighbor interaction, which we believe gives a more complete picture on the behavior of the spin liquid phase. We find that even a very small third-neighbor interaction can greatly stabilize the spin liquid regime.

## 5.2 Relevance to materials

This model has recently attracted much attention for its potential relevance to the material YbMgGaO$_4$. Experiments find enticing evidence for a spinon Fermi surface state from ther-

modynamic and inelastic neutron scattering measurements [9, 11]. Our work addressed the theoretical basis for such physics.

Our results support the claim of Ref. [20] that YbMgGaO$_4$ likely falls outside of the spin liquid phase in the presence of only first- and second-neighbor interactions. We found, however, that a very small third-neighbor interaction can greatly increase the size of the spin liquid phase and may appear quite naturally in the material. However, using the simple PSG picture, we always find that the Dirac spin liquid is energetically favored over the spinon Fermi surface state.

While the above results do not support the spinon Fermi surface state, we did find some effects which could tilt the balance in its favor. We saw that the spin-orbit interactions favor the spinon Fermi surface over the Dirac spin liquid state when we include effects beyond the simple projected mean-field wave functions. This leaves open the possibility that the spinon metal could be energetically favorable, perhaps assisted by other factors such as disorder or a small ring-exchange interaction.

If we assume that a spinon metal state does exist, interesting features emerge due to spin-orbit coupling. We showed how the spin-orbit interactions could explain the existence of broad peaks at the $M$ points in the spin structure factor and also predicted that measurements of the spin-polarized structure factors would display specific polarization-dependent peaks reflecting the anisotropic interactions. We also propose that the spin-orbit coupled spinon metal state may have a rather large thermal Hall conductivity which could be a very clear signature of spin liquid physics in such a system.

## 5.3 Future directions and implications

Looking forward, we anticipate a number of implications for the results and techniques developed in this work. For our spin-orbit coupled triangular systems, we showed that the restrictions imposed on the standard Gutzwiller-projected free fermion states by the PSG are quite severe for several of the $U(1)$ QSL states. Consequently, they are unable to adapt to strongly anisotropic interactions, and this may open the door to competition from $\mathbb{Z}_2$ QSL states in the case of such anisotropic models. In turn, this would be of considerable interest as the Gutzwiller-based approach almost always favors $U(1)$ states in Heisenberg models. The possibility of inducing fully gapped topological QSLs should be explored in the future by VMC techniques.

We argued that the thermal Hall effect should be a key signature of itinerant spinon excitations in spin-orbit coupled systems. While we obtained such an effect for the $U(1)$ spinon Fermi surface state on the triangular lattice, it was in fact suppressed by the PSG-mandated vanishing of effective spin-orbit coupling on the fermionic spinons at the free-particle level. Ultimately, this suppression owes itself to the presence of inversion symmetry, which, in conjunction with time-reversal symmetry, act on the spinons analogously to the way they do on real electrons. As is well known, the combination of inversion and time reversal in that context imply an exact two-fold Kramers degeneracy of the full electronic band structure, and a similar effect occurs here. When inversion is absent, for example, when an electric field is present normal to a two-dimensional electron gas, spin splitting occurs. The Rashba spin-orbit coupling induced by such a field is known to induce a large anomalous Hall effect in that context [33]. This strongly suggests that one should look for an enhanced thermal Hall effect in two-dimensional magnetic materials in which the magnetic layer has an asymmetric environment. This criteria, along with the requirement of large spin-orbit coupling, should assist in a search for this phenomenon.

Our methodology offers a consistent and quantitative method to compare QSLs and ordered phases for anisotropic magnetic Hamiltonians. This should have broad applicability to other materials such as the Kitaev compounds $\alpha$-RuCl$_3$, Na$_2$IrO$_3$, and Li$_2$IrO$_3$ in all its struc-

tural variations, and to three-dimensional systems like rare earth pyrochlores and spinels. The ability of VMC-based methods to tackle large systems is a unique numerical advantage. We expect many insights from such studies in the future.

# Acknowledgments

We thank Radu Coldea, Martin Mourigal, Gang Chen, and Yuan-Ming Lu for useful discussions. This work was supported by the NSF Materials Theory program through Grant No. DMR1506119 (J.I., C.L., L.B.) and by the Gordon and Betty Moore Foundation's EPiQS Initiative through Grant No. GBMF4304 (G.B.H.). We acknowledge support from the Center for Scientific Computing from the CNSI,MRL: an NSF MRSEC (DMR-1121053).

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
