# Peer review of "Spin Liquid versus Spin Orbit Coupling on the Triangular Lattice"

_SciPost Physics, doi:SciPost Phys. 4, 003 (2018)_

## Round 2 · Referee Report · Anonymous (Referee 1) · 2017-10-18

Strengths

Perform a systematic analysis of the spin model by using correlated wave functions. Detailed phase diagrams are reported.

Weaknesses

The statement in section 4 on the way to obtain a size extensive wave function is simply wrong.

Report

In this paper, the authors study a spin model by using variational wave functions and Monte Carlo methods. I think that the paper is nice and can be published. However, there is a huge mistake in section 4. Indeed, the claim that Eq.(16) gives a size extensive improvement in the calculation of observables is simply wrong. A size consistent improvement is only possible by retaining the exponential form of the projection operator, as in Eq.(15). Any expansion is not size extensive. In fact, Eq.(16) is EXACTLY what is done when people perform one Lanczos step:

|Psi_{1LS}> = (1+alpha H)|Psi_{VMC}>

Therefore, their approach IS equivalent to one Lanczos step. Naively, Eq.(16) may seem size extensive, but by considering the variation with respect to alpha, in order to minimize the energy, one can easily realize that alpha is not O(1/L) and therefore the energy gain is not size extensive.

Anyway, Eq.(16) is completely equivalent to performing one Lanczos step, as described in the references cited. Minor points:

1) In the introduction, the expression "one performs Monte Carlo sampling of the quantum wave function in real space" is sloppy. I suppose that this means that the sampling is performed in a (many-body) basis set where electrons/spin are localized on each site. "Real space" is reminiscent of a one-body sampling.

2) I suppose that both C_2 and S_6 in Eq.(2) also affect the spin operators. This should be mentioned. Moreover, a picture for the symmetry generators would help the reader. Is the S_6 symmetry a combination of a 120 rotation plus an inversion?

3) I do not find the definition of the vector n_{ij} in Eq.(3).

4) Just to clarify: on page 5, the number 18 (different mean-field ansatze) is referred to the Z_2 classification, correct?

5) At page 6, what do you mean by "all spin-dependent quadratic terms are prohibited"?

6) I suggest to introduce the extended model with J_2 and J_3 at the beginning. At present, J_2 is mentioned at page 8 without any reference to the actual Hamiltonian.

7) In the caption of Fig.3, what is the red region? I suppose the ordered 120-degree state, but this is not written. I am also disappointed by the layout of the figure: why not putting the label on the x axis? why showing the long arrow? It would be better to have 5 panels with labels a), b), c)... and no arrow. (The same for Fig.5).

8) I suppose that the value of the local moment reported at page 9 are for the simple Heisenberg model on the triangular lattice, but this is not written.

9) Finally, figures are too small and must be enlarged.

In summary, the paper contains a wrong statement, but nevertheless it contains relevant numerical calculations on an interesting model. Therefore, I ask for a revision; after that the paper can be published.

Requested changes

1) Delete or strongly correct section 4.

2) see my minor points in the report (especially enlarge figures).

  • validity: good
  • significance: good
  • originality: good
  • clarity: good
  • formatting: reasonable
  • grammar: perfect

Author:  Jason Iaconis  on 2017-10-31  [id 185]

(in reply to Report 1 on 2017-10-18)

We would like to thank the referee for their review of our manuscript and for recommending its publication. However, we believe that they have likely misunderstood key ideas of our method. The main point of emphasis in our method is that, instead of working directly with the modified wave function, we use the variational Monte Carlo to directly calculate the perturbative corrections to operator expectation values at a given order in $\alpha$. Critically, after expanding Eq.(15), these corrections can be found by calculating only the connected contributions to the expectation value. Note, that this is just a numerical application of the well known `Linked Cluster Theorem', which states that all contributions to the quantum correlation function can be found by summing only the connected Feynman diagrams. We use Eq.(16) as an intermediate step to obtain the numerical form of these connected contributions, which we describe in Eqs.(17) and (18).

Referee: "Any expansion is not size extensive. "

If one were to naively apply our Eq.(16), one would find such a sub-extensive correction after optimizing over $\alpha$. However, then one formally retains all powers of $n$ in the perturbative expansion and groups together powers of $\alpha$, the linked cluster theorem guarantees an exact cancelation of 'disconnected' terms in the numerator with the normalization factor in the denominator. Each term in the perturbative expansion is therefore given only by the connected part of the operator expansion, which scales linearly with system size, giving an extensive correction to all correlation functions. The linked cluster theorem is proven rigorously in standard many-body textbooks, and clearly results in an expansion of the partition function which is size extensive.

Referee: "In fact, Eq.(16) is EXACTLY what is done when people perform on Lanczos step: $|Psi_{1LS}\rangle = (1+ \alpha H)|Psi_{VMC} \rangle. $"

First of all, Eq.(16) is different from the result one gets using one Lanczos step. In that case, the last term in the numerator appearing at order $\alpha^2$, denoted as $\mathrm{Re} [ \langle \sf{H}^2 \mathcal{O} \rangle_0 ]$, would not be present. As has long been appreciated in perturbative QFT, it is critical to properly count the number of diagrams contributing at any order in a perturbative expansion in order to ensure an exact cancelation of vacuum diagrams. Additionally, we never actually apply Eq.(16) directly in our numerical simulation, but only use it as a starting point for further derivations. When calculating the correction to an observable, we expand both the numerator and the denominator in Eq.(16) and truncate our results at a finite order in $\alpha$. In fact, we could safely add terms of order $\alpha^3$ and further to Eq.(16) without changing any of our results.

Referee: " Therefore, their approach IS equivalent to one Lanczos step. Naively, Eq.(16) may seem size extensive, but by considering the variation with respect to alpha, in order to minimize the energy, one can easily realize that alpha is not O(1/L) and therefore the energy gain is not size extensive."

We reiterate, that we do not apply Eq.(16) directly, but expand this equation further to arrive at our final expressions. The resulting expressions for the corrections to general observables are given in Eq.(17) and to our specific choice of operators in Eq.(18). Notice how all these expressions scale with system size as $O(N)$, due to an exact cancelation of "vacuum diagrams".

Referee: "Anyway, Eq.(16) is completely equivalent to performing one Lanczos step, as described in the references cited."

As we have explained, Eq.(16) is both different from the result one finds by performing one Lanczos step and is only used as a starting point for further derivations in our numerical method. Furthermore, we do not work directly with the modified wave function, as is done when applying a single Lanczos step, but instead calculate the corrections to operator expectation values, using the connected correlation functions. To clarify this point, we have added the term ``$+O(\alpha^3)$'' to the numerator and denominator of Eq.(16), emphasizing that this equation is only used as an intermediate step in our derivation.

Referee: "Minor points: 1) In the introduction the expression ``one performs Monte Carlo sampling of the quantum wave function in real space" is sloppy. I suppose that this means that the sampling is performed in a (many-body) basis set where electrons/spins are localized on each site. "Real space" is reminiscent of a one-body sampling. "

We have reworded the corresponding statement for clarity.

Referee: "2) I suppose that both $C_2$ and $S_6$ in Eq.(2) also affect the spin operators. This should be mentioned. Moreover, a picture for the symmetry generators would help the reader. Is the $S_6$ symmetry a combination of a 120 rotation plus an inversion?"

We have added one more equation [Eq.(3)] where we describe how each symmetry operation transforms the spin components. We have also included a sentence above Eq.(2) where we clarify that the sixfold roto-reflection $\mathcal{S}_6$ is indeed ``a combination of a $120^\circ$ rotation and an inversion''.

"3) I do not find the definition of the vector $n_{ij}$ in Eq.(3)."

We have included an appropriate definition of $n_{ij}$ below Eq.~(4) [which used to be Eq.~(3)].

4) Just to clarify: on page 5, the number 18 (different mean-field ansatze) is referred to the Z-2 classification, correct?

Indeed, this is correct. To clarify this even further in the text, we now explicitly state that ``there are at least 18 different $\mathbb{Z}_2$ mean-field ansatze.

Referee: "At page 6, what do you mean by ``all spin-dependent quadratic terms are prohibited"?"

We mean that the PSG equations do not allow any hopping terms in the mean-field Hamiltonian which depend on the orientation of the spin [and therefore break SU(2) invariance]. To make this even more transparent, we have slightly reworded the corresponding sentence.

Referee: "6) I suggest to introduce the extended model with $J_2$ and $J_3$ at the beginning. At present, $J_2$ is mentioned at page 8 without any reference to the actual Hamiltonian."

We have added a line directing readers to the definition of the next-nearest neighbor term $J_2$.

Referee: "7) In the caption of Fig.3, what is the red region? I suppose the ordered 120-degree state, but this is not written. I am also disappointed by the layout of the figure: why not putting the label on the x axis? Why showing the long arrow? It would be better to have 5 panels with labels a), b), c)..., and no arrow. (The same for Fig.5).}"

We do not understand the referee's first question; we explicitly state it in the caption of Fig.~3 that the red region corresponds to the $120^\circ$ order. In accordance with the referee's further suggestions, we have added a label to the $J_{\pm\pm}$ axes of Figs.~3 and 5. For compactness, however, we have not separated each plot into different panels.

Referee: "8) I suppose that the value of the local moment reported at page 9 are for the simple Heisenberg model on the triangular lattice, but this is not written."

For clarity, we now state it explicitly that the corresponding DMRG results are ``for the triangular-lattice Heisenberg model''.

Referee: " 9) Finally, figures are too small and must be enlarged."

We have increased the sizes of our figures (where possible).

Referee: "In summary, the paper contains a wrong statement, but nevertheless it contains relevant numerical calculations on an interesting model. Therefore, I ask for a revision; after that the paper can be published."

As we have thoroughly addressed, the referenced statement is in fact correct. Since we have also addressed all of the referee's minor points by making minor revisions in the manuscript, we believe that it can be published in its current form.

---

## Round 2 · Referee Report · Anonymous (Referee 2) · 2017-12-15

Strengths

1- The model studied has rich phase behaviour and is relevant to actual materials, 2- The method used is adequate and encompasses the state of the art analytical and numerical techniques, 3- The results obtained are very interesting and relevant to the experiments.

Weaknesses

Non

Report

The quantum spin models with bond-dependent anisotropic spin interactions have recently become one of the most popular topics
among the physicist working in the field of quantum magnetism. In such models, anisotropy driven spin frustration can give rise to a number of interesting phenomena, exotic ordered states and quantum spin liquid. Such anisotropic interactions are inherent to Mott insulators with strong spin-orbit coupling.

The authors study the generic anisotropic quantum spin model, with all symmetry allowed terms, on a triangular lattice.
Based on combined analytical and numerical techniques, the authors present an exhaustive study of rich phase behavior of the model, its spin correlations, emergent fermionic spectrum and unusual Hall transport.

The methods employed are justified, obtained result are new and very interesting and are relevant to Mott insulators of heavy elements in general and to the triangular lattice spin system YbMgGaO4 in particular.

The paper is very well written and the comments of the first reviewer have been fully reflected in the revised version.

I do recommend the paper for the publication in its present form.

Requested changes

Non

---

## Round 3 · Referee Report · Anonymous · 2017-11-14

Report
I thank the authors for their clarifications on section 4. Indeed, I was confused because of the symbol of an expectation value was used in the left hand side of Eq.(16).
Within the Lanczos step procedure (which is not size consistent), a real expectation value is evaluated.
Instead, here, the right hand side of Eq.(16) (which I believe is size consistent) does not correspond to an expectation value.
In this regard, the variational principle is lost and it is not clear whether the optimization procedure is stable and meaningful.
Is there always a finite value of alpha for which a variational energy is obtained?
I do not have particular problems with that, however, I would ask the authors to make a precise remark on this fact in the paper.
Moreover, I ask the authors to remove the $\langle ... \rangle$ symbol in Eq.(16) (and related equations) and use a different notation.
Author: Jason Iaconis on 2017-11-27 [id 196]
(in reply to Report 1 on 2017-11-14)
We would once again like to thank the referee for their review of our manuscript. We are happy to have clarified the point of confusion and again thank the referee for recommending its publication.
While the right hand side of Eq.(16) is still an expectation value, it is true that it is not directly measured in our VMC simulation. To make this point clear, we have taken the referee's suggestion and used a separate notation to differentiate between the theoretical operator which we are approximating with our perturbative expansion and expectation values which we measure directly in the VMC simulation. We have added a sentence to the manuscript explaining this distinction. We have also added several sentences clarifying that our method is only exact in the perturbative limit and may lead to results which are no longer variational for finite alpha.
Intended Changes:
1) We have changed the left hand side of Eq's (15) and (16) from $\langle \mathcal{O} \rangle$ to $\langle \langle \mathcal{O} \rangle \rangle$.
2) We have added the following sentences on Pg. 13:
"We use the symbol $\langle \langle \cdots \rangle \rangle$ to differentiate the theoretical operator expectation values in Eq's~(15) and (16) from expectation values we evaluate directly within the VMC simulation."
"Note that Eq.~(16) can not be measured directly within the VMC simulation, but instead can be measured approximately by expanding perturbatively in $\alpha$. As such, the results of our calculation are only variational up to the accuracy of the asymptotic expansion. "
As we have addressed the referee's remaining concerns, we believe the updated manuscript can now be published in its present form.
Anonymous on 2017-11-11 [id 188]
Indeed, I must confess that I misunderstood their approach for the Lanczos step procedure.
I was confused by the expectation value on the left side of Eq.(16). Instead, the right side of this equation does not correspond to any expectation value over a variational wave function (in contrast to the Lanczos step technique).
This fact implies that the variational property for the energy is lost and the optimisation may lead to antivariational results.
I have nothing against this approach, but I ask the authors to add a clear remark on this property, possibly removing the symbol <O> from the left side of Eq.(16), and all subsequent equations.

---

## Round 3 · Author Response

We would like to thank the referee for their review of our manuscript and for recommending its publication. However, we believe that they have likely misunderstood key ideas of our method. The main point of emphasis in our method is that, instead of working directly with the modified wave function, we use the variational Monte Carlo to directly calculate the perturbative corrections to operator expectation values at a given order in $\alpha$. Critically, after expanding Eq.(15), these corrections can be found by calculating only the connected contributions to the expectation value. Note, that this is just a numerical application of the well known `Linked Cluster Theorem', which states that all contributions to the quantum correlation function can be found by summing only the connected Feynman diagrams. We use Eq.(16) as an intermediate step to obtain the numerical form of these connected contributions, which we describe in Eqs.(17) and (18).
Referee: "Any expansion is not size extensive. "
If one were to naively apply our Eq.(16), one would find such a sub-extensive correction after optimizing over α. However, then one formally retains all powers of n in the perturbative expansion and groups together powers of $\alpha$, the linked cluster theorem guarantees an exact cancelation of 'disconnected' terms in the numerator with the normalization factor in the denominator. Each term in the perturbative expansion is therefore given only by the connected part of the operator expansion, which scales linearly with system size, giving an extensive correction to all correlation functions. The linked cluster theorem is proven rigorously in standard many-body textbooks, and clearly results in an expansion of the partition function which is size extensive.
Referee: "In fact, Eq.(16) is EXACTLY what is done when people perform on Lanczos step: |Psi1LS>=(1+ $\alpha$ H)|PsiVMC>."
First of all, Eq.(16) is different from the result one gets using one Lanczos step. In that case, the last term in the numerator appearing at order $\alpha^2$, denoted as $\Re[⟨H^2 O⟩_0]$, would not be present. As has long been appreciated in perturbative QFT, it is critical to properly count the number of diagrams contributing at any order in a perturbative expansion in order to ensure an exact cancelation of vacuum diagrams. Additionally, we never actually apply Eq.(16) directly in our numerical simulation, but only use it as a starting point for further derivations. When calculating the correction to an observable, we expand both the numerator and the denominator in Eq.(16) and truncate our results at a finite order in $\alpha$. In fact, we could safely add terms of order $\alpha^3$ and further to Eq.(16) without changing any of our results.
Referee: " Therefore, their approach IS equivalent to one Lanczos step. Naively, Eq.(16) may seem size extensive, but by considering the variation with respect to alpha, in order to minimize the energy, one can easily realize that alpha is not O(1/L) and therefore the energy gain is not size extensive."
We reiterate, that we do not apply Eq.(16) directly, but expand this equation further to arrive at our final expressions. The resulting expressions for the corrections to general observables are given in Eq.(17) and to our specific choice of operators in Eq.(18). Notice how all these expressions scale with system size as O(N), due to an exact cancelation of "vacuum diagrams".

---

## Round 3 · List of Changes

Referee: "Anyway, Eq.(16) is completely equivalent to performing one Lanczos step, as described in the references cited."
> As we have explained, Eq.(16) is both different from the result one finds by performing one Lanczos step and is only used as a starting point for further derivations in our numerical method. Furthermore, we do not work directly with the modified wave function, as is done when applying a single Lanczos step, but instead calculate the corrections to operator expectation values, using the connected correlation functions. To clarify this point, we have added the term "+O($\alpha$^3)'' to the numerator and denominator of Eq.(16), emphasizing that this equation is only used as an intermediate step in our derivation.
Referee: "Minor points:
1) In the introduction the expression one performs Monte Carlo sampling of the quantum wave function in real space" is sloppy. I suppose that this means that the sampling is performed in a (many-body) basis set where electrons/spins are localized on each site. "Real space" is reminiscent of a one-body sampling. "
> We have reworded the corresponding statement for clarity.
Referee: "2) I suppose that both C2 and S6 in Eq.(2) also affect the spin operators. This should be mentioned. Moreover, a picture for the symmetry generators would help the reader. Is the S6 symmetry a combination of a 120 rotation plus an inversion?"
> We have added one more equation [Eq.(3)] where we describe how each symmetry operation transforms the spin components. We have also included a sentence above Eq.(2) where we clarify that the sixfold roto-reflection S6 is indeed a combination of a 120 rotation and an inversion''.
"3) I do not find the definition of the vector nij in Eq.(3)."
> We have included an appropriate definition of $n_{ij}$ below Eq.~(4) [which used to be Eq.~(3)].
>4) Just to clarify: on page 5, the number 18 (different mean-field ansatze) is referred to the Z-2 classification, correct?
>Indeed, this is correct. To clarify this even further in the text, we now explicitly state that there are at least 18 different Z2 mean-field ansatze.
Referee: "At page 6, what do you mean by all spin-dependent quadratic terms are prohibited"?"
>We mean that the PSG equations do not allow any hopping terms in the mean-field Hamiltonian which depend on the orientation of the spin [and therefore break SU(2) invariance]. To make this even more transparent, we have slightly reworded the corresponding sentence.
Referee: "6) I suggest to introduce the extended model with J2 and J3 at the beginning. At present, J2 is mentioned at page 8 without any reference to the actual Hamiltonian."
> We have added a line directing readers to the definition of the next-nearest neighbor term J2.
Referee: "7) In the caption of Fig.3, what is the red region? I suppose the ordered 120-degree state, but this is not written. I am also disappointed by the layout of the figure: why not putting the label on the x axis? Why showing the long arrow? It would be better to have 5 panels with labels a), b), c)..., and no arrow. (The same for Fig.5).}"
> We do not understand the referee's first question; we explicitly state it in the caption of Fig.~3 that the red region corresponds to the 120' order. In accordance with the referee's further suggestions, we have added a label to the J±± axes of Figs.~3 and 5. For compactness, however, we have not separated each plot into different panels.
Referee: "8) I suppose that the value of the local moment reported at page 9 are for the simple Heisenberg model on the triangular lattice, but this is not written."
>For clarity, we now state it explicitly that the corresponding DMRG results are for the triangular-lattice Heisenberg model''.
Referee: " 9) Finally, figures are too small and must be enlarged."
>We have increased the sizes of our figures (where possible).
Referee: "In summary, the paper contains a wrong statement, but nevertheless it contains relevant numerical calculations on an interesting model. Therefore, I ask for a revision; after that the paper can be published."
> As we have thoroughly addressed, the referenced statement is in fact correct. Since we have also addressed all of the referee's minor points by making minor revisions in the manuscript, we believe that it can be published in its current form."

---

## Editorial Decision

published